# Experience is central and connections matter: A Leximancer analysis of the research priorities of people with lived experience of mental health issues in Australia

**Michelle Banfield** [1,2]*, **Amelia Gulliver**[1,2], **Dana Jazayeri**[1,3], **Victoria J. Palmer**[1,3], **the ALIVE National Centre for Mental Health Research Translation Investigator Group**[1]

1 The ALIVE National Centre for Mental Health Research Translation, The University of Melbourne, Melbourne, Victoria, Australia, 2 Centre for Mental Health Research, The Australian National University, Canberra, Australian Capital Territory, Australia, 3 The Department of General Practice, Melbourne Medical School, Faculty of Medicine, Dentistry and Health Sciences, The University of Melbourne, Melbourne, Victoria, Australia

* Michelle.Banfield@anu.edu.au

## Abstract

Mental health research priority-setting has a long history internationally. Many of these studies use expert panels or consensus methods across multiple mental health stakeholder groups. Whilst such approaches are designed to produce agreed research priorities, there is a risk that the specific and nuanced priorities of less powerful groups, especially those with lived experience of mental health issues, are lost in translation. We aimed to develop Australian mental health research priorities from the perspectives of people living with mental ill-health and their carers, families and kinship group members. A cross-sectional, open-ended survey was conducted nationally in Australia during January and February 2022. We asked participants to list three priorities on which mental health research should focus. Responses were analysed using Leximancer, a text analytics tool, to examine the concepts and their connections across the data. A total of 365 people with lived experience of mental ill-health participated in the survey. The majority (57%) identified as consumers, with 14% identifying as carers and 29% reporting both types of lived experience. Participants were from all Australian states and territories and from metropolitan, regional and remote areas. The Leximancer analysis generated 30 concepts in six thematic priority areas. The most prominent themes were experience, treatment and impact, followed by stigma, peer and trauma. The concept maps displayed complex connections and interrelationships between specific concepts, with lived experience a large and central concept. Analysis of the textual responses emphasised the importance of examining specifics, as the nuanced research priorities traversed themes and concept across the maps. This project provides robust evidence on the central importance of experience as driving mental health lived experience research priorities. Further, it demonstrates that people with lived experience describe the key issues in complex, interrelated ways that require multi-factorial research approaches to address.

**Data Availability Statement:** The full dataset is available from the ANU Data Commons, doi: 10. 25911/j2dy-p787 A searchable database of the dataset used in the current study is available at https://alivenetwork.com.au/mental-health-priorities/.

**Funding:** The ALIVE National Centre for Mental Health Research Translation is supported by a grant from the National Health and Medical Research Council (GNT2002047) awarded to VJP. The funding body played no role in study design, data collection, analysis, decision to publish or preparation of the manuscript.

**Competing interests:** The authors have declared that no competing interests exist.

## Introduction

Establishing priorities for mental health research and translation has been the focus of research for several decades. Studies have investigated either broad ranging priorities for mental health research, or priorities for specific groups such as children and young people or certain issues such as eating disorders and suicide prevention [1]. On a global scale, efforts to establish consensus among different groups such as service providers, government policy makers, research experts and more recently with people with lived experience of mental ill-health and/or carers/family and kinship group members has been considerable [1–5]. Previous papers that have examined research priorities of service users, experts by experience or 'consumers and carers' (Australian government terms referring to people with lived experience of mental ill-health and carer/family and kinship group members), have illustrated that priorities between different groups may be shared; however, they can also be distinctive and occasionally in direct conflict [4, 6, 7]. In some cases, how a participant values certain priorities may be dependent upon individual factors, such as the perceived impact of specific funded mental health programs [7]. In other cases, topics may be shared priority areas such as medication, but the specific priorities of focus within these areas are distinctive. Nevertheless, despite much previous research in this field, overall priorities for mental health research and translation have remained similar over time, focusing on integration of services, the role of social care and support, a need for a peer-led models, recovery, and addressing the impacts of medication [1].

Limited evolution in priority-setting is problematic, potentially representing redundant efforts, or a potential lack of progress in mental health research implementation and translation [1]. A clear example of this issue can be shown by examining a recent Lancet Psychiatry paper [8] that aimed to establish actionable, evidence-based priorities for preventing earlier loss of life related to mental illness and distress. The study was an action from an international S-Plan road-mapping effort with a working group of experts, including people with lived experience. Eighteen actionable areas were identified, across the areas of "integration of mental health physical healthcare", "prioritisation of prevention while strengthening treatment", and "optimisation of strengthening synergies across social-ecological levels and the intervention cycle" [8]. Solutions such as "eliminate silos in healthcare", "work-based interventions to promote healthy workplaces" and "increase investment in mental health" both reflect and reinforce the decades of existing priority setting exercises and in the case of the latter, lobbying, but offer little in the way of new and specific areas for action. What we lack are the multi-modal and multi-factorial models for implementation to address the well-established mechanisms to target.

Another critical issue is establishing whose priorities are being reflected, particularly within large scale priority-setting exercises. While people with lived experience may be included within larger exercises [e.g., 4, 5], the subjectively nuanced priorities of what matters for people with lived experience is frequently lost in translation. For example, higher order themes are generated from original data, but without preserving participants' language, eroding their intended meaning and communicating a lived experience voice "second-hand" [1, 9]. In addition, few funding schemes have clear links between the allocation of resources back to the priorities of people with lived experience, and instead are often developed based on burden of disease indicators [10]. Thus, it is important to not only centre the priorities of those most impacted directly, but to ensure the priorities also reflect their language, goals and desires to ensure that we get the solutions right.

### The ALIVE national centre for mental health research translation roadmap

Australia's first nationally focused Mental Health Research Translation Centre (called the 'ALIVE National Centre' herein) was funded by the Australian National Health and Medical Research Council (NHMRC) as part of a special initiative in mental health (2021–2026). The special initiative was established to tackle the intractable mental health service delivery, system and experience issues that have resulted in large-scale inequities in health outcomes, a significant and unacceptable continued gap in life years related to unmet physical health needs - particularly for priority populations such as Aboriginal and Torres Strait Islander people and people who live with severe mental ill-health - and to implement new models of care and innovations to address these issues. ALIVE National Centre investigators span 17 Australian universities, are interdisciplinary (across health and medical disciplines, social science, the creative arts, humanities and health services research, and health economics), with founding partner organisations responsible for mental health service delivery, clinical innovation in health care, mental health policy advocacy and translational research. A research governance framework encompasses centre activities with three co-directors sharing responsibilities for decision-making, and lived experience co-chairs (people living with mental ill-health and carer, family and kinship group members) appointed across an independent advisory board, a policy and practice committee, and international scientific committee.

A blueprint was developed in 2021 during the ALIVE National Centre establishment and detailed several interrelated research activities that continued over 2022 [11]. The blueprint activities include: a narrative review and synthesis [1]; a policy scan to establish the state of play in regard to mental health research translation within national mental health and suicide prevention government plans, frameworks and strategies; ecosystem mapping to establish a picture of place-based initiatives relevant to the ALIVE National Centre's research objectives of prevention across the life course, unmet physical health needs in priority populations and lived experience led-research and research priorities within these areas; an annual, national co-partnered lived experience priorities survey to continuously gather and update priorities; and prioritisation using public co-design methods with people with lived experience and carers/family and kinship group members to develop phased Consensus Statements on implementation actions for priority areas. These materials form a dynamic, living national roadmap for mental health research translation. Phased Consensus Statements on the Implementation Actions as articulated through surveys and public co-design with people with lived experience can be found on our website [12].

The current paper reports on the 2022 annual lived experience priorities survey conducted as part of the blueprint activities for the co-design of the national roadmap. Building on our previous priority-setting work [6], the aim of this study was to develop updated Australian mental health research priorities from the perspectives of people living with mental ill-health and their carers, families and kinship group members.

## Methods

### Ethics approval

The ethical aspects of the study were approved by The Australian National University Human Research Ethics Committee (amendment to Protocol number 2013/388). All participants provided written informed consent to participate.

## Lived experience involvement

This research was led by lived experience researchers, who designed the study, collected and analysed the data, and authored the paper. Others with lived experience, including members of the Consumer and Carer Advisory Group associated with ACACIA: The ACT Consumer and Carer Mental Health Research Unit, and ALIVE National Centre Co-Design Living Labs Co-Leads assisted with the design. Our public co-design prioritisation process and development of the Consensus Statements provided additional opportunity for lived experience involvement in analysis, interpretation and the development of implementation actions.

## Study design

A cross-sectional, open-ended survey was designed in 2021–22. This survey was based on previous priority-setting exercises conducted by lived experience researchers at the Australian National University (ANU), for ACACIA: The ACT Consumer and Carer Mental Health Research Unit [6]. It was developed by ALIVE National Centre lived experience and co-design researchers with input from members of the ACACIA Advisory Group and Co-Design Living Labs Co-Leads. The online survey was hosted on the ANU Qualtrics platform; the link was distributed widely through Australian mental health networks as described below, with a suggested completion time of 20 to 30 minutes.

## Participants and recruitment

For consistency with the prior work and in recognition of language about mental health and lived experience being in transition in Australia, we chose to use the terms "consumer and carer" in recruitment and data collection. Individuals residing in Australia over 18 years of age, who self-identified as either a consumer, carer or as both consumer and carer were invited to complete the survey. In order to self-select speaking positions, people were provided with the following definitions of consumer and carer/family members:

> "A consumer is someone with lived experience of mental ill-health, whether or not they have accessed services."

> "A carer is a family member or friend who provides informal (i.e. unpaid) support to a mental health consumer."

We note within the field that some definitions of consumer restrict definitions to only those who have accessed a service for mental health issues. In this case the definition was intentionally broad to acknowledge and include diverse experiences, including recovery and work in the suicide prevention space, to formulate our national research agenda and strategies to address intractable problems.

As an exploratory and generative study, sample size and power calculations were not appropriate. To ensure saturation, detailed distribution lists were formulated by the research teams. These included links sent out via email and using the social media channels LinkedIn and Twitter. Invitations were sent to existing members of the ACACIA Register, which is a list of people who have an interest in contributing to mental health research, and posted to the ACACIA Facebook page. A wider list for distribution was created for further targeted invitations, comprising consumer and carer state and territory peak bodies and known networks, and organisations with an interest in mental health research translation who had connections with people with lived experience and the establishment of mental health research priorities. Organisations were invited to share the link to the survey within their email updates, newsletters or

to forward to membership lists where appropriate. There were over 200 invitations distributed to organisations and individuals. Recruitment commenced on 14 January 2022 and the survey was closed on 9 March 2022.

## Data collection

The participant information sheet was included within Qualtrics. Participants were required to acknowledge reading the information sheet and provide informed consent by answering yes to the following statement prior to survey completion: *"I have read and understood the information about this research project. I understand that any information I provide from this point onwards will be included in the research, even if I don't finish the survey."*

The survey comprised basic demographic information (age, gender and postcode, which was used by the team to establish state of residence and urban, regional or remote contributions); characteristics of their lived experience (identification as a consumer, carer or both consumer and carer; how they chose to describe their mental health e.g., diagnosis) and included two questions specifically to support identification of priorities and priority-setting processes within the ALIVE National Centre. The first question asked people to *'share three things mental health research should focus on'* with the statement accompanying this that "there is no right or wrong answer here to this question just what is important to you". People were given three boxes for responses: Priority 1, Priority 2 and Priority 3. There was no intention to gather these as a hierarchically formulated list from people or conduct numerical ranking on the importance of these. The second question asked people to share their preferred methods for research involvement of people with lived experience; responses are the focus of a separate paper [13]. At the end of the survey, people were invited to opt-in to hear more about the online prioritisation to complete the additional blueprint activity (also reported separately, forthcoming).

Participants were able to complete the survey in several sessions if preferred: cookies enabled return to a partially-completed survey within one week of commencement, after which surveys were automatically submitted. Data from partially completed surveys were included in analyses. Qualtrics features to prevent multiple completions by the same individual and bot detection were also enabled. Measures to uphold anonymity included not asking for personal details (for example, names and email addresses), deletion of IP address immediately upon download of the data from Qualtrics, and re-coding of postcode level details into larger state and territory codes. Postcodes were also used to establish remoteness according to the Modified Monash Model (MMM), from MM1 (Metropolitan) through to MM7 (Very remote communities) [14]. The survey data were stored on a secure ANU server with access only by those named on the project. De-identified data were used for the Leximancer analysis.

## Data analysis

Leximancer is "a text analytics tool that can be used to analyse the content of collections of textual documents and to display the extracted information visually" [15]. Leximancer creates concept maps for words that travel together within text-based documents and these concepts indicate some element of being interrelated or connected. A concept is formed based on a cluster of words that travel together frequently in the text. A theme is a way of organising these concepts to reflect that they hold relationships within the text. This means that concepts that are closely related all sit under the same theme and grey lines between concepts show how they are interrelated or connected in some way with other concepts. A concept may sometimes have the same title as a theme (for example, 'impact' can be a concept that sits within a theme named 'Impact') reflecting the conceptual content of the data and the suggested theme by

Leximancer. The proximity of concepts to each other and different sizes of their associated circles, represent frequency of use within the text. The larger the circle, the more prominent the concept within the text. Sometimes if concept circles are very large, this suggests the concept is over-connected within the text. Over-connection of a concept can lead to a loss of meaning and the suggestion is to remove the concept due to what the Leximancer manual terms being "bleached of meaning."

Similarly, the colours used for maps follow a colour wheel and a heat mapping approach. Here, hotter colours, for example red, represent the concepts that are most discussed in text followed by brown, orange, green, blue to purple.

Several analytical stages were completed (Fig 1). As per the Leximancer procedures for data analysis, an initial .csv file was uploaded to the software program and a preliminary analysis was run to produce the first Leximancer generated map (see S1 File). The first map provided an overview of the data where one can see most themes and the interrelated concepts, and also allowed assessment of overly large concepts (bleached concepts) and words like "think," "maybe," "perhaps" or other words that represent what are called stopping words (words used in our general speech patterns that represent a pause or thinking word or the end of a sentence). As detailed in S1 File, the Leximancer software concept seed editor list function was used for the removal of (1) bleached concepts, (2) stopping words, and (3) any words contained in the survey or interview questions so that these were not included in the concept analysis. Here, the terms "mental" "health" "research" and "three" "things" were excluded. During these iterations, singular and plural concepts were also merged. Following this, a number of map iterations were completed to arrive at the final stage map, including a textual analysis to further examine the meanings suggested by people about the generated concepts and the interrelationships of these as priorities for mental health research.

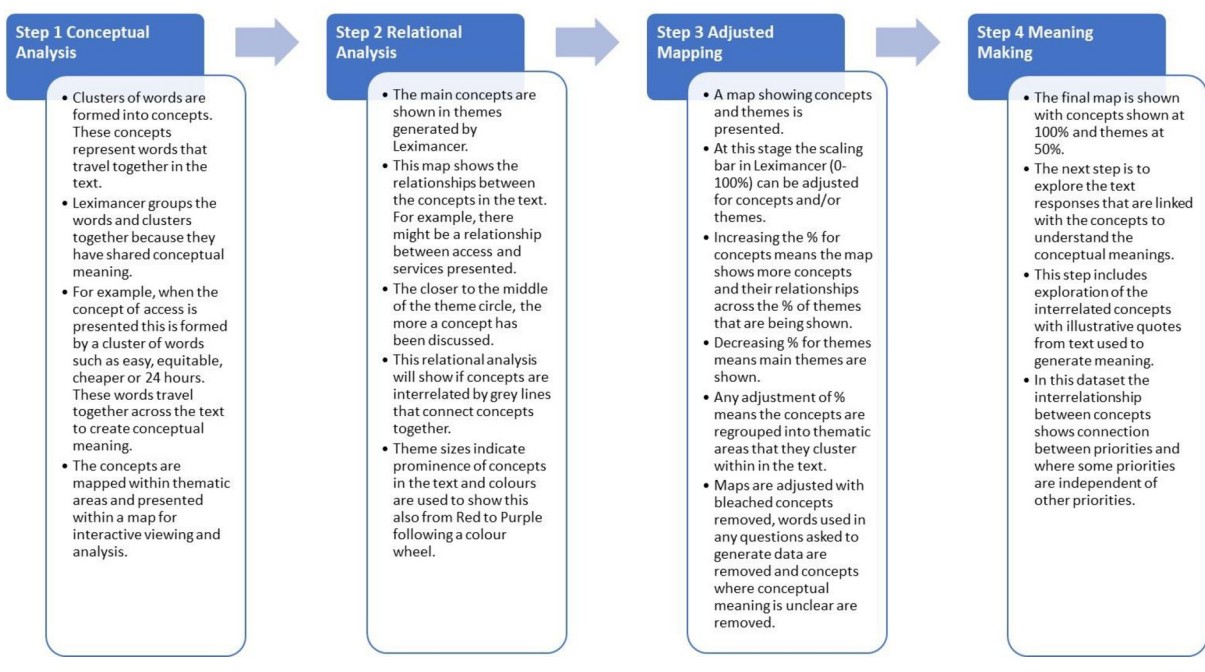

**Fig 1. Leximancer process and map stages of analysis.**

## Results

### Participant characteristics

By closing date of the survey there were 510 responses. Of these, one person declined consent, 26 did not identify as someone with lived experience, and 118 did not list any research priorities. A total of 365 people who identified as either consumers, carers, or with experience of being both consumer and carer contributed at least one research priority. There were some instances where people had listed more than one research priority within an individual text box response: these instances were reviewed, and the priorities were separated out for inclusion in the analysis. The result was 1,294 research priorities shared by people in response to the request "Please share three things that mental health research should focus on." As shown in Table 1, of the 365 contributors, 207 (57%) identified as being consumers, 52 (14%) as carers and 106 (29%) as both consumer and carer. As such, in presenting illustrative quotes there may be a weighting in favour of responses from consumers and people who identified as both.

Participants were invited to share how they described their own mental health, or the mental health of the person they care for, in their own terms. The majority of participants listed

**Table 1. Participant characteristics.**

| Role | N (n = 365) | % |
|---|---|---|
| Consumer | 207 | 57% |
| Consumer-Carer | 106 | 29% |
| Carer/Family | 52 | 14% |
| **Age** | Mean = 46.5, SD = 14.5, range = 20–93 (n = 343*) | |
| **Gender** | | |
| Female | 280 | 77% |
| Male | 65 | 18% |
| Another term | 17 | 4% |
| Prefer not to say | 3 | 1% |
| **State** | | |
| Australian Capital Territory | 18 | 5% |
| New South Wales | 79 | 22% |
| Northern Territory | 9 | 3% |
| Queensland | 47 | 13% |
| South Australia | 23 | 6% |
| Tasmania | 11 | 3% |
| Victoria | 134 | 37% |
| Western Australia | 38 | 10% |
| *Not reported* | *6* | *1%* |
| **MMM remoteness category** | | |
| MM1 (Metropolitan) | 242 | 66% |
| MM2 (Regional centres) | 38 | 10% |
| MM3 (Large rural towns) | 11 | 3% |
| MM4 (Medium rural towns) | 12 | 3% |
| MM5 (Small rural towns) | 43 | 12% |
| MM6 (Remote communities) | 2 | <1% |
| MM7 (Very remote communities) | 3 | <1% |
| *Not reported* | *14* | *4%* |

* 22 participants did not provide an age

more than one mental health condition or descriptor. Responses included diagnostic terms such as depression, anxiety, complex post-traumatic stress disorder, borderline personality disorder, obsessive compulsive disorder, bipolar disorder, schizophrenia and schizoaffective disorder; many respondents also listed other terms such as stress, grief, family violence and suicidality, or chose to use descriptors such as "complex," "unmet long term [needs]" and "exhausting".

The age range for all respondents was from 20 and 93 years old, and most respondents were female (n = 280, 77%). There were responses from people across all states and territories of Australia, though over half of respondents were located in Victoria or New South Wales. Two-thirds of respondents were from metropolitan areas (MM1), but all MMM remoteness categories were represented.

### The first stage–mapping the priorities of people with lived experience analysis

Fig 2 shows the final Leximancer map for concepts and themes as grouped by Leximancer analysis. Following the heat mapping approach, the prominent themes and concepts within the text were: *experience* (red), *treatment* (brown) *impact* (green), *stigma* (blue/green), *peer* (blue) and *trauma* (purple). In Fig 2 it is also possible to see how the themes (priority areas) and concepts (priorities) overlap. For example, concepts talked about within *experience* overlap with *treatment*, such as community, family and appropriate. However, other *experience* concepts may be independent of *treatment* (such as education, therapy and access to services). Concepts of stigma and impact are also interconnected with experience. These indicate that underlying mechanisms for what needs to be targeted include an interrelationship with experience.

In Fig 3, the interconnected and interrelated concepts are shown without the Leximancer-generated themes and by following the grey lines it is possible to interpret the textual responses further from the map.

Journeying across the conceptual grey lines made it possible to determine which concepts were interconnected and which were interrelated in the textual responses. For the purposes of this analysis, the differences between interconnected and interrelated concepts might be best understood as: interconnected concepts representing distinctive priorities, compared with interrelated concepts that showed priorities across many different aspects of the textual responses. Interconnected concepts typically appeared as offshoots from a main concept (indicated by size) and might end after one or two concepts were shown together. For example, from the central concept of lived experience, it was possible to establish the connection between 'lived experience and education' and 'lived experience and community'. The positioning of these concepts near to lived experience on the map is important as it showed that these priorities mattered for people in their responses as related to experience; however, being interconnected means these were concepts that were independent and distinctive of others in the text. These independent and distinctive priority areas were also seen in the connection between 'experience and therapy and access and services.' Here therapy, access and services were not discussed in relation with other concepts and thus these can be said to have reflected distinctive priorities for people once again for mental health research.

Within the overlapping areas of *stigma* and *impact*, system appears as an independent research priority connected with work, but looking at the grey line we can see that this discussion about work was also interrelated with concepts of distress and stigma. Thus, from a view point only of the map and not considering the textual responses that were given, it is possible to interpret that work may be associated with distress and stigma for people with lived experience in ways that make this an important priority for research.

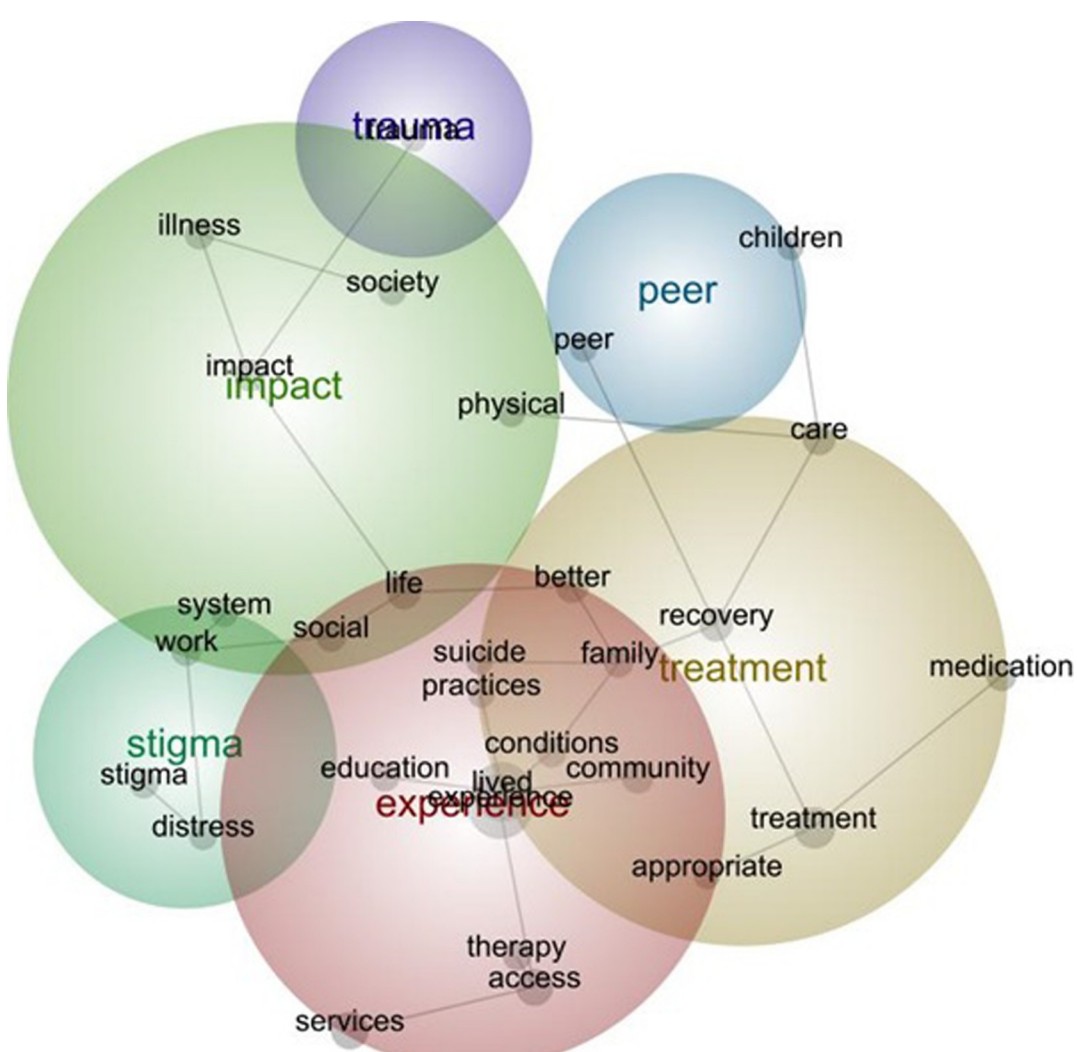

**Fig 2. Final Leximancer map of the six priority areas (themes) and the priorities (concepts) shared by people with lived experience as important for mental health research.**

Travelling back up the grey line from work, there is an interrelationship shown between stigma and work with social, life and impact. Again, building on the story just presented, this indicated that work, distress and stigma were concepts of importance that may also be interrelated with the concepts of better and family and recovery. However, it is important to highlight that system sits independently, connected to work but not interrelated with social, life and impact. Again, this indicated that conceptually stigma and work and the interaction with the concept of system were priorities for people, but these were distinct from the ways that work, social, life and stigma, and impact were discussed.

Another core interrelated concept for impact is trauma, which is positioned as connected but separate from impact. Hence, it is possible that from a viewpoint of the map that trauma is a distinct priority area from the interrelated concepts of the impact of illness and society.

As final examples to illustrate these distinctions between how interconnected and interrelated concepts may be interpreted further, it is possible to see that treatment is connected with concepts of appropriate and medication, and the concept of care is connected with distinctive

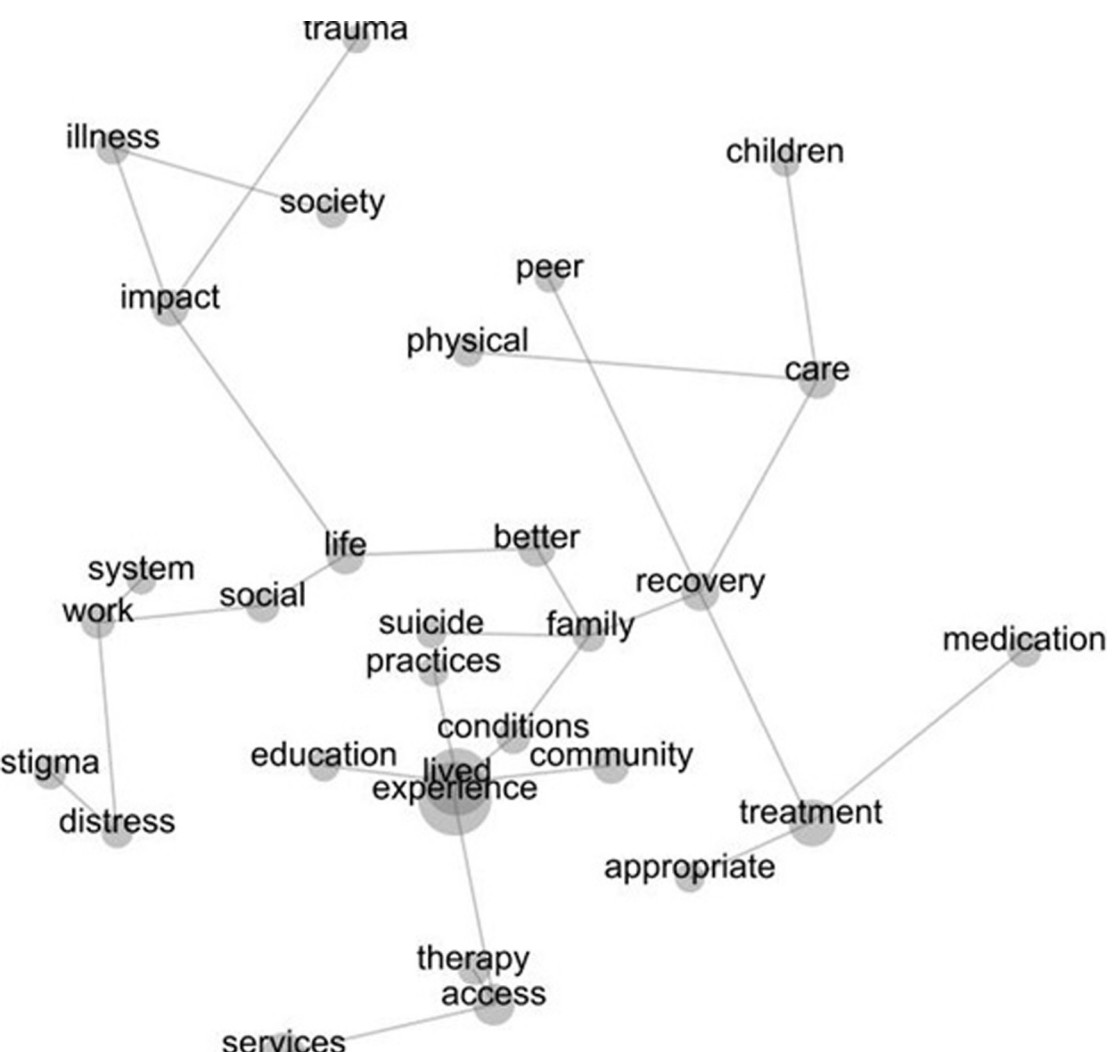

**Fig 3. Final Leximancer generated concept map of the important priorities for mental health research shared by people with lived experience.**

priorities for physical [health care], and children. The connected but separate nature of these concepts indicated distinctive priorities for research on each of these topics. This contrasts with the concepts related with recovery. Recovery is a priority, but is conceptually understood with relation to family, suicide, practices and lived experience. However, while unsurprisingly treatment and peers are interrelated, with peers separated from other concepts this showed that this remains a distinctive research priority for people.

### From a conceptual point of view to understanding the meanings underpinning the priorities for people with lived experience

As described in the overview of the maps generated by the Leximancer analysis, the themes that showed the conceptual groupings of interconnected and interrelated concepts indicated six main priority areas for research for consumers, carers and consumer/carers: *experience*, *treatment*, *impact*, *stigma*, *peer/s* and *trauma*. For a deeper understanding of the meaning and connection than suggested by the maps alone, it is necessary to explore how participants

phrased their priorities. The source of each quote below is identified as from a consumer, carer or someone with both experiences, followed by their reported gender and age where available. The full list of priorities as shared by participants is available in a searchable database on our digital platform [16], including some basic demographics and tags to allow searching by groups and topics.

One example to illustrate diversity of interrelated meanings is in further consideration of how *experience* and lived experience were discussed in responses. One person's response was to research, "recovery experiences and progress after a critical incident" (Consumer, Female, 71), and another referred to research priorities as "sharing experiences for understanding [their] condition" (Consumer, Female, 41). However, in many cases, research priorities extended to connected concepts and intersections: "experiences of people with intersectional experiences in terms of mental health. Inclusive of situations where people have suicidal ideation (in intense pain) but no plan" (Consumer, Female, 45); "intersection of sexual violence and mental ill-health" (Consumer, Female, 27). Each of these examples illustrate the central importance of experience as a mental health research priority, but with the final two demonstrating how experience also connected with other concepts in this area.

The importance of lived experience as a strongly interrelated research priority was further illustrated in the intersection of experience and access to services, as people described "managing the knife edge walk between lived experience strength-based self-image and the change resistant pathology filter to access services" (Both, Female, 70); and "the qualitative evidence of experience and support services" (Consumer, Male, 30). People's responses also demonstrated the complex interrelationships across multiple themes and concepts, including traversing *experience*, *impact* and *peer* to prioritise "greater workplace and access needs for people with lived experience" (Both, Female, 53) and needing to "compare services delivered within lived experience workforces versus services without a lived experience workforce." (Consumer, Male, no age). These examples illustrate that when priorities are reduced to short thematic statements or priority areas combined for ranking, we risk overlooking the nuanced connections and meanings that people with lived experience see as centrally important.

This issue was also one that emerged in the context of *treatment* as a priority area. For example, when discussing treatment and experiences, some people referred to simply wanting better, more effective, responsive or non-pharmacological options. However, there was also reference to more specific issues, such as to focus on "research into long term side effects of medication which could lead to better medication" (Consumer, Female, 47), and the importance of "long term treatment. Not just six funded consultations and then see you later" (Both, Female, 72). As for *experience*, responses also demonstrated the connections between concepts and themes shown in the maps, with responses that traversed many interrelated concepts such as "the potential for combining mental health and chronic pain treatment for more holistic management" (Both, Female, 28); "effectiveness of various treatments for trauma" (Carer, Female, 74); and "the importance of family and carers in the support plan was important as families and carers are cut out and the focus is all on the younger person which disconnects them further from the people who care for them" (Both, Female, 55). Treatment and recovery were also conceptually interrelated, but this reflected a holistic view of recovery that encompassed social issues. People described "consumer participation in planning and the impacts on recovery" (Consumer, Female, 55), the "societal factors that impact recovery for example housing stability, poverty and discrimination" (Both, Female, 29) and prioritised "the impact of social prescribing on mental health recovery" (Consumer, Female, 58).

Some priorities in *treatment* also connected with concepts of care and physical [health], but as illustrated by the maps, these connections tended to reflect more distinct individual priorities. People suggested that there needed to be "more emphasis on physical health" (Both, Male,

80), research into "the effects of treating mental and physical health as different entities" (Both, Female, 55), and "affordable physical activity, nutrition, sleep, employment support that is designed for people living with mental illness" (Consumer, Female, 44). Interrelated concepts focused on the complexities of having co-occurring conditions, including "trauma and co-morbid physical health issues" (Consumer, Prefer Not to Say, 26), with one person reflecting on the difficulties of access to care: "how to get help for mental illness when you also have a physical disability and your care needs are too high for a psychiatric admission and you can't get help for psychiatric problems on a general ward of the hospital" (Consumer, Female, 40).

Within the *impact* theme, there were some priorities that were standalone concepts, including "how poor clinical workplace culture impacts on mental health consumers and what needs to change" (Consumer, Male, 27), "the impact of psychiatric and psy professions and coercion, paternalism and violence" (Both, Female. 49), and "how communication around mental health/illness impacts those with chronic/complex mental health, for example, lots of discussion around depression is focused on episodic depression, which can alienate those with chronic depression or mood disorders." (Consumer, Use another term, 30). However, many responses reflected strong reciprocal relationships between concepts. One example was to explore "mental health and work and the relationship between and impact of each on one another" (Both, Female, 28). Reference was also made to research prioritising "the mental health peer workforce in Australia and its impact and effectiveness" (Consumer, Male, 26) and "societal impacts" (Carer, Male, 54). Other areas of intersecting impacts included financial impacts, lack of resources, poverty, gendered impacts on mental health, the impact of COVID on kids and young people (children in the map), ways to prevent isolation, impacting changes to child protections policies, and the "mental health impact for carers" (Carer, Female, 44). The link between *impact* and *trauma* was particularly salient, with people suggesting "prevention of long lasting impacts from trauma" (Consumer, Female, 61), "understanding how trauma causes mental ill-health" (Consumer, Female, 27), the "influence of trauma on diagnosed mental illnesses" (Consumer, Prefer Not to Say, 26) and research that explains "the effects of trauma" (Consumer, Female, 50) and "childhood trauma" (Both, Female, 65) were important priorities.

In relation to *stigma* people referred to the need to remove, ease, reduce, lessen and challenge stigma as distinct priorities. However, the interrelations of concepts in this theme that are illustrated in the maps were reflected in the nuanced ways people suggested research should contribute to these actions. People were particularly interested in how experiencing stigma affects people, and suggested that mental health research should focus on "how did you feel about the stigma applied to you." (Consumer, Female, 46), and should "explain the distress of experiencing stigma in the mental health system and work place" (Consumer, Female, 50). Of note was the view that it was key to "[research] the stigma of mental health in the community without resorting to separating people without one, and [be] inclusive rather than exclusive" (Consumer, Male, No Age).

For *peer* the "connection and belonging" (Consumer, Female, 37) provided by peer groups was important, and the use of peer workers for "breaking the poverty cycle" (Both, Female, 46) was an additional distinct research priority. Consumers also referred to the importance of implementation of peer workers for case management and for named conditions such as psychosis, with some noting that they would specifically like research to prioritise "how lived experience workforce impacts people at early stage of diagnoses" (Consumer, Female, 33). Research priorities also reflected a desire for research to explicitly explore roles, value, impact and positioning within systems. People suggested research into "the role and value of peer workers" (Both, Female, 52), "peer led models and the impact it makes" (Both, Female, 30) and "what forms of lived experience involvement actually affect practice (in clinical and policy

settings)? What structures and avenues of lived experience involvement make the biggest impact and show engagement substantively rather than tokenistically" (Both, Male, 28). Thus, although this theme was amongst the smallest and lowest in prominence on the heat map, there was still significant complexity to the underlying priorities.

## Discussion

The findings of this study emphasise the centrality of experience in consumer and carer priorities for research, but also demonstrate the complexity of the way priorities are phrased and the connections between them to form nuanced topics of focus. The six priority areas largely confirm core areas that have been in focus in research for decades, with some advance made in determining where priorities may be interrelated or interconnected. Experience, treatment, and stigma have been recurrent priority areas shown in countless priority-setting exercises [3, 17, 18]. Trauma has likewise been a priority area that has grown in its prominence in the past decade [6, 19]. Our own work has also demonstrated the importance of research into the peer workforce [6, 7], although this was an area that did not feature in work conducted prior to 2010 [20, 21], and the peer workforce was conspicuously low key in O'Connor and colleagues recent healthcare solutions for addressing premature mortality [8].

Some may argue that this persistence of themes may be attributed to a lack of knowledge of existing research by people with lived experience, as there are large bodies of research in many of these areas. However, although research may appear to be in the right areas, the repetition of the priorities suggests our approaches are not working. As we first observed more than a decade ago [21], an alternate explanation is that research and translation are failing to address the specific and often intricately connected priorities of the people who should be the ultimate beneficiaries of research and its translation. In the earlier study, whilst there was some alignment between broad priority areas, when the nuanced topics developed by people with lived experience were matched against published research, there was little alignment [21]. As participants in the current study stated, first-hand accounts from people with lived experience are critical for understanding where systems, including research, are failing, and there is a need to listen to lived experience voices as they "challenge and disrupt how academics and funders think" [13].

What is noteworthy from our Leximancer analysis is how it has mapped out the connections between priorities, and particularly drawn out the conceptual importance of impact - impacts of trauma, impacts of stigma - and the central conceptual framing of lived experience. Most specific priorities, encompassing services, family, physical health, illness, social and work life are interrelated within experience and impact, reflecting a desire for research to approach the salient issues with this interrelated nature front of mind, focused on the impacts for those affected. This speaks to the need to tailor interventions based on an understanding of these interrelationships and ensure that implementation addresses the complex intersecting factors to create a holistic approach. This would progress the conclusions of the Lancet Psychiatry Commission blueprint for protecting physical health in people with mental illness, which suggested that a syndemic approach and integrated models that were also customised to settings (including social settings) may improve prevention and interventions [22].

We noted in the introduction that the issue of meaning being lost in translation within research efforts has been identified in our recent narrative review and synthesis of the priorities for mental health research and translation over the past twelve years [1]. This loss of meaning is an example of the epistemic injustice that is common in mental health care and research [9, 23]. Part of the threat to the salient meanings of those most impacted lies in efforts of large-scale studies [e.g., 4, 5, 8] to reach consensus both within and across various groups with interest. A

focus on consensus carries the risk that genuinely diverse views and complex meaning are lost, and trying to achieve this across groups with unequal power in priority-setting will silence some voices, favouring clinical, academic and professional knowledge over lived experience [9, 23]. This risk is acknowledged in these other priority-setting studies, with explicit efforts to elevate lived experience voices amongst the many. However, in the case of the Gone Too Soon study, the provided definition of lived experience suggests that they may have counted the views of "those whose jobs or service bring them into direct engagement with those who are suicidal or have a mental illness" within lived and living experience [8], thus further blurring the boundaries of which voices are central and reinforcing hierarchies of knowledge [9].

Our own approach to these issues has been to conduct priority-setting exclusively with people with lived experience [6, 7, 20, 21], building on this earlier work with the ALIVE National Centre annual survey as reported in this paper. Rather than seeking rankings or forcing priorities into higher order categories with expert panels, in the ALIVE National Centre we use public co-design to bring together survey contributions to co-develop our Consensus Statements, which maintain the issues, suggested actions and societal factors as described by people with lived experience. This focus on creating our roadmap for mental health research from lived experience priorities reflects our embedded model of lived experience, which builds our research agenda and practice from the views of those most impacted. Using our co-designed roadmap, research conducted within the ALIVE National Centre is expected to demonstrate alignment with these lived experience priorities; how to effect change through research and translation then becomes a collaborative endeavour that also privileges lived experience perspectives to address enduring issues in an integrated and holistic manner.

A specific co-design process led by Aboriginal and Torres Strait Islander researchers is underway to establish First Nations community-led priorities as a pathway within the roadmap. It is anticipated that multiple pathways will be needed across different priority populations to reflect the diversity of communities and lived experience across Australia and to elevate priorities of groups where inequities remain high, unaddressed and unacceptable. In 2023, work commenced on a families' pathway with the annual priorities survey focused on parents living with mental ill-health, partners/caregivers of people living with mental ill-health, and children living with families of a parent experiencing mental ill-health. The ALIVE National Centre's roadmap will be implemented over the duration of the Centre funding (2021–2026) and an impact framework will be co-created to evaluate progress annually and update priorities in the national roadmap. To extend this beyond our own work we have made a searchable database of complete participant responses to the annual surveys freely available on our digital platform [16] to encourage those in research and translation beyond our networks to engage directly with the priorities as they were written. This allows experiential knowledge to be central to our entire mental health research, care and policy system to address the marginalisation perpetuated by epistemic injustice [23].

## Limitations and future directions

Our choice to distribute the invitation to participate via mental health organisations and channels may have affected the nature and scope of participant views. Although we achieved a good distribution of participants across demographic characteristics, as is frequently the case in mental health research, the majority of participants identified as female, and were from metropolitan areas in the most populous states of Australia. Further work is needed to engage with people who identify as male or use another term, to explore whether priorities for these groups differ in content and/or emphasis. Likewise, specific investigation of research priorities for those in regional, rural and remote areas is required to ensure that the unique challenges faced

in these areas are in focus for mental health research; this work is planned for our 2024 annual survey. Due to the size and complexity of the dataset, we chose to combine responses across consumer and carer expertise for the current analysis; however, as our previous research has demonstrated some important differences in these views, further analysis by type of lived experience is warranted.

## Conclusions

Our aim in this study was to update the mental health research agenda from the perspectives of people with lived experience of mental ill-health and their carers, family and kinship group members. Harnessing the unique text analytics of Leximancer allowed us to explore not only individual priorities, but how they naturally grouped into priority areas and most importantly, connected with each other to reflect the meanings central to participants. It is not surprising that lived experience is the central theme that connects all others. However, what is critical for researchers and those involved in translation to understand and act upon is the importance of the impacts of the distinct and interrelated priorities. The themes and concepts illustrated in the maps reflect the depth and complexity of the issues people with lived experience want research to address, and the priorities as written by the participants themselves offer tremendous scope for research with impact. The challenge for researchers is to resist the temptation to translate these to continue "business as usual", and instead accept the invitation of people with lived experience to follow the new paths they have set.

## Supporting information

**S1 File. Leximancer analysis stages and maps.**
(DOCX)

**S2 File. ALIVE mental health priority-setting survey.**
(DOCX)

## Acknowledgments

We thank the participants who generously shared so many detailed ideas for research priorities. We also acknowledge the contributions of the broader ALIVE National Centre team in designing the survey and managing recruitment, Laura Hemming for preparing the data for analysis and conducting trial Leximancer map generations, and Elton Lobo for the programming of the shared priorities database.

## Author Contributions

**Conceptualization:** Michelle Banfield, Victoria J. Palmer.

**Data curation:** Michelle Banfield, Amelia Gulliver, Dana Jazayeri, Victoria J. Palmer.

**Formal analysis:** Michelle Banfield, Amelia Gulliver, Victoria J. Palmer.

**Funding acquisition:** Victoria J. Palmer.

**Methodology:** Michelle Banfield, Victoria J. Palmer.

**Project administration:** Michelle Banfield, Dana Jazayeri, Victoria J. Palmer.

**Writing – original draft:** Michelle Banfield, Amelia Gulliver, Victoria J. Palmer.

**Writing – review & editing:** Michelle Banfield, Amelia Gulliver, Dana Jazayeri, Victoria J. Palmer.

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
