## [Decision Letter · Decision Letter 0]

13 Feb 2024

PMEN-D-24-00020

Experience is central and connections matter: A Leximancer analysis of the research priorities of people with lived experience of mental health issues in Australia

PLOS Mental Health

Dear Dr. Banfield,

Thank you for submitting your manuscript to PLOS Mental Health. After careful consideration, we feel that it has merit but does not fully meet PLOS Mental Health’s publication criteria as it currently stands. Therefore, we invite you to submit a revised version of the manuscript that addresses the points raised during the review process.

We look forward to receiving your revised manuscript.

Kind regards,

Vitalii Klymchuk, Ph.D., D.Sc.

Academic Editor

PLOS Mental Health

Journal Requirements:

1. In the online submission form, you indicated that "A searchable database of the dataset used in the current study is available at https://alivenetwork.com.au/mental-health-priorities/. The full dataset is available from the corresponding author on reasonable request.". 

3. Uploaded as supplementary information.

2. Please send a completed 'Competing Interests' statement, including any COIs declared by your co-authors. If you have no competing interests to declare, please state "The authors have declared that no competing interests exist". 

3. Please provide separate figure files in .tif or .eps format.

https://journals.plos.org/mentalhealth/s/figures 

https://journals.plos.org/mentalhealth/s/figures#loc-file-requirements 

Additional Editor Comments (if provided):

Reviewers' comments:

Reviewer's Responses to Questions

**Comments to the Author**

1. Does this manuscript meet PLOS Mental Health’s publication criteria? Is the manuscript technically sound, and do the data support the conclusions? The manuscript must describe methodologically and ethically rigorous research with conclusions that are appropriately drawn based on the data presented.

Reviewer #1: Yes

Reviewer #2: Yes

2. Has the statistical analysis been performed appropriately and rigorously?

Reviewer #1: Yes

Reviewer #2: N/A

3. Have the authors made all data underlying the findings in their manuscript fully available (please refer to the Data Availability Statement at the start of the manuscript PDF file)?

Reviewer #1: Yes

Reviewer #2: Yes

4. Is the manuscript presented in an intelligible fashion and written in standard English?

Reviewer #1: Yes

Reviewer #2: Yes

5. Review Comments to the Author

Reviewer #1: Thank you for the opportunity to review this interesting paper. It is well considered and logically presented, focusing on an important topic. The description of the methodological processes are rigorous and it is clear the steps that the research team took in analysing the data and its final presentation.

The final figures were a little hard to read, but the descriptions accompanying them were clear.

It is particularly important that this project takes a lived experience priority position in developing the research priorities, and this is balanced well with the views of carers. I did have a question concerning the design of the overall methodology - was this something that was also co-designed with the relevant community? This is a little unclear.

I also wondered whether those with lived experience contributed to the research overall - from design through to data collection and onto analysis. Was the community consulted for example in the final themes put forward in the analysis? If not this could be an opportunity for further study, along with understanding whether consumers and carers differed in their priorities.

Reviewer #2: Notes

- The roles of all the authors to be added in as per recent guidelines on collaborations.

- Same protocol number (perhaps generic) was found (Protocol number 2013/388) in a similar article by the same author - Mental health research priorities in Australia: a consumer and carer agenda

- Broader literature review might help - a good number of the references were self-citations - not bad in itself but limited to finding research in a lager context.

Other notes

- Good identification of socioeconomic factors as limitation

- Interesting read.

6. PLOS authors have the option to publish the peer review history of their article (what does this mean?). If published, this will include your full peer review and any attached files.

**Do you want your identity to be public for this peer review?** For information about this choice, including consent withdrawal, please see our Privacy Policy.

Reviewer #1: No

Reviewer #2: No

---

## [Editor Report · Decision Letter 1]

13 Mar 2024

Experience is central and connections matter: A Leximancer analysis of the research priorities of people with lived experience of mental health issues in Australia

PMEN-D-24-00020R1

Dear Prof Banfield,

We are pleased to inform you that your manuscript 'Experience is central and connections matter: A Leximancer analysis of the research priorities of people with lived experience of mental health issues in Australia' has been provisionally accepted for publication in PLOS Mental Health.

Before your manuscript can be formally accepted you will need to complete some formatting changes.

Please also ensure you address the following:

In the methods please provide a sample size and power calculation or describe how participant recruitment was designed to ensure saturation.

-In the Discussion/Conclusions – in the limitations, please consider adding a point about whether the way in which the survey invitation was distributed (through networks of people/agencies interested in MH research) might affect the results. Could also mention here whether there are any concerns about saturation if you did not perform a priori sample size calculation

-Please provide a copy of the survey as a Supporting Information file.

-Please confirm that there are no restrictions around publishing Leximancer data

Best regards,

Vitalii Klymchuk, Ph.D., D.Sc.

Academic Editor

PLOS Mental Health